# Using CSF Proteomics to Investigate Herpesvirus Infections of the Central Nervous System

**DOI:** 10.3390/v14122757

**Published:** 2022-12-10

**Authors:** Saima Ahmed, Patrick van Zalm, Emily A. Rudmann, Michael Leone, Kiana Keller, John A. Branda, Judith Steen, Shibani S. Mukerji, Hanno Steen

**Affiliations:** 1Department of Pathology, Boston Children’s Hospital, Harvard Medical School, Boston, MA 02115, USA; 2Neuroimmunology and Neuro-Infectious Diseases Division, Department of Neurology, Massachusetts General Hospital, Harvard Medical School, Boston, MA 02115, USA; 3Department of Pathology, Massachusetts General Hospital, Harvard Medical School, Boston, MA 02115, USA; 4F.M. Kirby Neurobiology Center, Boston Children’s Hospital, Harvard Medical School, Boston, MA 02115, USA; 5Precision Vaccines Program and Neurobiology Program, Boston Children’s Hospital, Harvard Medical School, Boston, MA 02115, USA

**Keywords:** liquid chromatography/mass spectrometry (LC/MS), CSF proteomics, herpesvirus, biomarkers, CNS, human herpes simplex 1 (HSV-1), human herpes simplex virus 2 (HSV-2), varicella zoster virus (VZV human herpes 3)

## Abstract

Herpesviruses have complex mechanisms enabling infection of the human CNS and evasion of the immune system, allowing for indefinite latency in the host. Herpesvirus infections can cause severe complications of the central nervous system (CNS). Here, we provide a novel characterization of cerebrospinal fluid (CSF) proteomes from patients with meningitis or encephalitis caused by human herpes simplex virus 1 (HSV-1), which is the most prevalent human herpesvirus associated with the most severe morbidity. The CSF proteome was compared with those from patients with meningitis or encephalitis due to human herpes simplex virus 2 (HSV-2) or varicella-zoster virus (VZV, also known as human herpesvirus 3) infections. Virus-specific differences in CSF proteomes, most notably elevated 14-3-3 family proteins and calprotectin (i.e., S100-A8 and S100-A9), were observed in HSV-1 compared to HSV-2 and VZV samples, while metabolic pathways related to cellular and small molecule metabolism were downregulated in HSV-1 infection. Our analyses show the feasibility of developing CNS proteomic signatures of the host response in alpha herpes infections, which is paramount for targeted studies investigating the pathophysiology driving virus-associated neurological disorders, developing biomarkers of morbidity, and generating personalized therapeutic strategies.

## 1. Introduction

Herpes simplex viruses 1 and 2 (HSV-1 and HSV-2) and varicella-zoster virus (VZV) are human neurotropic DNA viruses with an ability to establish latency in the nucleus and both innate and adaptive immune systems are required to limit reactivation and central nervous system (CNS) entry throughout the lifetime of the host [1,2]. These three viruses are members of the alpha herpesviruses subfamily, which are characterized by short reproductive cycles, cellular destruction, and latency in sensory nerve ganglia. Location of primary dormancy and neurological consequence of reactivation in the host differs by herpesvirus type. HSV-1 is largely dormant in the trigeminal ganglion, while HSV-2 inhabits the sacral sensory ganglia, and VZV is latent in the dorsal root ganglion; all three herpesviruses can cause meningitis, encephalitis, or meningoencephalitis and disease severity is estimated to be due to viral and host-related factors [1,3,4].

While metagenomic next-generation sequencing (mNGS), in comparison to traditional diagnostic workflows, is at the forefront of capturing pathogen-specific genetic material for rapid and accurate diagnosis [5] host response in herpes infection at the proteomic level is understudied. As these viruses are common causes of disease in immunocompromised patients because of their ability to reactivate when host immune responses fail, shedding light onto the immune and other biological pathway responses after infection is required for our understanding of disease heterogeneity and clinical outcomes. Characterizing the host response to a known infection at the proteomic level can provide valuable information to understand host physiology contributing to the disease process and identify potential targets for host-informed therapies. One such example is the neuroinflammation from HSV-1 infection, which is associated with the severity of neurodegenerative diseases, such as Alzheimer’s Disease (AD) and Parkinson’s Disease (PD) [6].

Mass spectrometry (MS)-based proteomics has become increasingly integrated into translational and clinical research. The ability to process tens to hundreds (or even thousands) of patient samples in a high-throughput manner is now standard in proteomic biomarker discovery workflows [7,8,9]. Proteomes of a wide range of human biofluid specimens, including CSF, have been successfully characterized using liquid chromatography–mass spectrometry (LC/MS)-based proteomics [7,8,10,11,12].

More specifically, coupling high-throughput, low-volume sample processing using just microliter amounts of human biofluids with state-of-the-art high-resolution mass spectrometers can be a powerful combination for the discovery of protein biomarkers and to study pathophysiological mechanisms of disease.

A comprehensive understanding of the biological pathways activated after HSV-1, HSV-2, and VZV CNS infection in real-world settings remain unexplored. Our overarching aim of this pilot study was to demonstrate the feasibility of our unique proteomics method by mapping CSF proteomes of patients infected with HSV-1 in comparison to those from patients infected with HSV-2 or VZV. We used HSV-1 as the starting point given its prevalence and frequent association with the most severe morbidity [13]. Here, we demonstrate the applicability of our CSF proteomics pipeline to the discovery of novel CSF biomarkers and biological processes contributing to the pathophysiology of herpesviruses in the context of CNS infection. We identify distinct proteomic signatures for HSV-1 in comparison to HSV-2 and VZV in the context of CNS infection, which could help provide mechanistic insight into comparative differences in host responses to counteract herpesvirus reactivation, help guide the discovery of effective vaccines against the different herpesviruses, and aid in additional therapeutic development for herpesviruses.

## 2. Materials and Methods

### 2.1. Patient Samples

Patient samples were obtained as part of a prospective cohort study enrolling adults who presented to Massachusetts General Brigham (MGB) with neurological symptoms concerning for neurological infection (fever, altered mental status, neck stiffness, seizure, and electroencephalographic or neuroimaging findings concerning infection) and had a lumbar puncture. Participants consented to the secondary use of samples to investigate pathogenicity and mechanisms of disease after neurological infection or inflammation. For this study, we mapped the CSF proteomes of patients in this cohort with confirmed infection by herpes simplex virus 1 (HSV-1), herpes simplex virus 2 (HSV-2), or varicella zoster virus (VZV), which we refer to as the herpesvirus group. Reasons for lumbar puncture, past medical history, and diagnostics used to confirm microbial pathogen are documented in Table 1. No participants were tested for 14-3-3 or real-time quaking-induced conversion (RT-QuIC). Since CJD testing precludes laboratories from performing other testing until CJD is excluded, providers do not routinely test unless the clinical scenario is consistent with the disease. The study was approved by the Partners Institutional Review Board under protocol 2015P001388.

### 2.2. Sample Processing Method

Our in-house-developed “MStern Blotting” approach was used to process the CSF samples [7]. In short, to 100 µL CSF, we added dithiothreitol (DTT) and (1:1 (*w*/*w*) urea in 1 M Tris/HCl pH 8.5 to a final concentration of 10 mM DTT. The resulting denatured and reduced CSF was incubated for 20 min at 27 °C and 1100 rpm in a ThermoMixer (Eppendorf, MA, USA). Reduced cysteine side chains were alkylated with 50 mM iodoacetamide (IAA) and incubated for 20 min at 27 °C and 750 rpm on a ThermoMixer.

The 96-well hydrophobic PVDF membrane plate (Millipore, MA, USA) was primed with 150 μL 70% ethanol and equilibrated with 300 μL 8 M urea supernatant and vacuumed through a 96-well plate-adaptable vacuum manifold (Millipore, MA, USA). All subsequent liquid transfers were conducted using this 96-well microplate vacuum manifold (Millipore, MA, USA). Each sample was vacuumed three times through the PVDF membrane 96-well plate. After adsorption of the proteins onto the membrane, proteins were washed twice with 50 mM ammonium bicarbonate. Protein digestion was performed using one μg of sequencing grade trypsin (Promega and 100 μL digestion buffer (5% acetonitrile ACN, 50 mM ABC and trypsin) were added to each well. After incubation for 2 h at 37 °C in a humidified incubator, the cleaved proteins (now peptides) were eluted through the vacuum onto a collection 96-well plate. Recovered peptides were eluted using a vacuum twice with 150 μL of aqueous 40% ACN containing 0.1% formic acid. Subsequently, the elution solutions were pooled and dried in a vacuum concentrator. Lyophilized samples were resuspended in 20 μL of MS loading buffer (5% FA in 5%ACN) before LC/MS analysis.

### 2.3. LC/MS-based CSF Proteome Mapping

CSF samples were analyzed using a PicoChip column (150 μm × 10 cm Acquity BEH C18 1.7 μm 130 Å, New Objective, MA, USA) with 2 μL min solvent A (0.1% FA) using a micro-autosampler AS2 and nanoflow HPLC pump module (Eksigent/Sciex, Framingham, USA). Proteolytic peptides were eluted from the column using 2% solvent B (0.1% FA in ACN) in solvent A, which was increased from 2–30% on a 40 min ramp gradient and to 35% on a 5 min ramp gradient at a flowrate of 1000 nL/min. The PicoChip, containing an emitter for nanospray ionization, was kept at 50 °C and mounted directly at the inlet to a Q Exactive Mass Spectrometer (Thermo Scientific, Bremen, Germany). The mass spectrometer was operated in positive DDA top 10 mode with the following MS1 scan settings: mass-to-charge (*m*/*z*) range from 375–1400, resolution 70,000 @ m/z 200, AGC target 3e6, max IT 60 ms. MS2 scan settings were: resolution 17,500 @ m/z 200, AGC target 1e5, max IT 100 ms, isolation window *m*/*z* 1.6, NCE 27, underfill ratio 1% (intensity threshold 1e4), charge state exclusion unassigned, 1, >6, peptide match preferred, exclude isotopes on, dynamic exclusion 40 s.

### 2.4. LC/MS Data Analysis

DDA runs were searched together using MaxQuant v 1.6.0.1 (Cox, Hein et al. 2014). Standard settings were used in MaxQuant with the following modifications: carbamidomethylated cysteine residues (fixed), acetylation of the N-terminal of proteins (variable), oxidation of methionine (variable). Match-between-runs analysis was enabled and only the filtered 1% FDR identifications were used. All statistical analysis was performed using R Studio, Perseus v 1.6.10.43, GraphPad Prism 9 (Dotmatics, MA, USA). The Protein–Protein Interaction Network was created using string-db.org [14] (Szklarczyk, Gable et al. 2019).

## 3. Results

### 3.1. Characteristics of CSF Proteomics Study Population

A total of thirteen CSF samples (seven male, six female; median age (interquartile range): 74 years (25–87)) were processed in a single batch (Figure 1). We compared three hospitalized patients with HSV-1, four patients with HSV-2, and six patients with VZV (Table 1). The median CSF white blood cell count for HSV-1, HSV-2, and VZV was 166.00 (136.75, 570.50), 164.25 (67.50, 308.88), and 169.50 (98.38, 215.75) and days from hospitalization to lumbar puncture was 1.33 (0.58), 0.75(0.96), 1.00(1.26), respectively. All were receiving treatment with acyclovir at the time of lumbar puncture.

### 3.2. Distinguishing the CSF Proteomes of Patients with HSV-1, HSV-2, and VZV

Our CSF proteomic pipeline identified a total of 1231 proteins. Our first question was to determine differential proteins expressed in these three herpes viral infections, focusing on comparisons with HSV-1 relative to HSV-2 and VZV. To achieve this, we first examined the proteomes using a pairwise *t*-test for the following contrasts: HSV-1 vs. HSV-2 and HSV-1 vs. VZV (Figure 2), followed by a bioinformatic analysis of the differentially regulated proteins using STRING to look for enriched biological pathways and related interacting proteins (Figure 3). We identified 70 proteins that showed significantly different abundances between HSV-1 and HSV-2 using a *p*-value cutoff of 0.05 (i.e., equivalent to −log_10_ of 1.3). Surprisingly, we identified a greater number of downregulated than upregulated proteins in HSV1 infections: 54 vs. 16 proteins (Figure 2a). While upregulated biological pathways in HSV-1 compared to HSV-2 infection primarily consisted of catabolic processes including the organic substance catabolic process (FDR 0.012) and cellular catabolic process (FDR 0.028) (Figure 3a), there was an overall downregulation in the cellular metabolism including the organic substance metabolic process (FDR 4 × 10^−4^) cellular nitrogen compound metabolic process (FDR 3 × 10^−4^), and small molecule metabolic process (FDR 2.2 × 10^−3^) in HSV-1 compared to HSV-2 infection. Other downregulated biological pathways were enriched in viral transcription (FDR 0.012) and translational initiation (FDR 4 × 10^−4^) (Figure 3b).

Comparison of HSV-1 and VZV infection showed a similar number of significantly upregulated and downregulated proteins in HSV1 infections (38 vs. 40) (Figure 2b). Among the 40 downregulated proteins, there were no significant pathway enrichment. In contrast, the proteins upregulated in HSV1 infections were associated with pathways such as regulation of apoptotic (FDR 4 × 10^−5^), immune system (FDR 8.7 × 10^−3^), and viral processes (FDR 3.1 × 10^−3^), as well as neurogenesis (FDR 1.5 × 10^−2^) and cytokine-mediated signaling (FDR 6.1 × 10^−3^).

Motivated by the significant differences found when comparing HSV1 with HSV2 or VZV, we next sought to perform a 3-way comparison using ANOVA to find significantly changing proteins between each herpes subtype. Our ANOVA analysis with a *p*-value cutoff of 0.05 found 78 significant proteins. Some of the most statistically significant proteins from the 78 ANOVA significant proteins distinguishing our three herpesviruses infections were Glutathione S-transferase omega-1 (GSTO1) (*p* = 4 × 10^−4^), 60S ribosomal protein L34 (RL34) (*p* = 1 × 10^−4^), 14-3-3 protein gamma (1433G) (*p* = 1 × 10^−4^), 14-3-3 protein epsilon (1433E) (*p* = 5 × 10^−3^), 14-3-3 protein eta (1433F) (*p* = 4 × 10^−3^), apolipoprotein B-100 (APOB) (*p* = 1.04 × 10^−2^), protein S100-A9 (S10A9) (*p* = 1.20 × 10^−3^), and protein S100-A8 (S10A8) (*p* = 1.50 × 10^−3^) (Figure 4a–h).

Next, we performed further bioinformatic analysis with hierarchical clustering of the 78 ANOVA-significant proteins differentiating our three herpesviruses to find the protein signatures for each of the three neuroinvasive infections (Figure 5). Our analysis revealed three virus-specific sub-clusters. We performed pathway analysis using STRING on each subcluster of upregulated proteins in HSV-1, HSV-2, and VZV, respectively. A sub-cluster of 21 proteins differentiated HSV-1 infection from other herpes infections and were related to biological pathways, such as chemokine production (FDR 1.20 × 10^−2^), inflammation-induced leukocyte migration (FDR 2.70 × 10^−2^), and apoptotic signaling pathway regulation (FDR 4.00 × 10^−4^). In the HSV-2 sub-cluster, 31 of 37 specifically upregulated proteins were associated with pathways related to metabolism (FDR 4.00 × 10^−4^), consistent with the pairwise analysis detailed above where we had noted an upregulation of metabolic processes when comparing HSV-2 to HSV-1. The VZV sub-cluster of 19 proteins did not reveal any significant pathways.

## 4. Discussion

In our study, we could identify proteomic differences in CSF of patients infected with one of three different herpesviruses with suspected involvement of the central nervous system (CNS). The three herpesviruses of interest were HSV-1, HSV-2, and VZV. Using a pipeline that can perform untargeted protein screening in minutes, we were able to identify over 1200 proteins in CSF, suggesting the feasibility of using convenience samples to identify proteins and pathways affected in infectious disease on a timescale that could have clinical benefit. In this study, we found 78 significantly altered proteins and associated biological pathways across the three most common herpesviruses that cause meningitis or encephalitis in adults and were able to detect biological differences when performing pairwise comparisons with each infection. These data are a proof of principle that proteomic analyses may help further our understanding of biological pathways involved in herpes viral reactivation and has the potential to assist in differentiating host responses to alpha herpesviruses.

In a comparison of HSV-1 to HSV-2, we found notable downregulation of metabolic and upregulation of catabolic processes in HSV-1 infection-associated host proteomes. Metabolic proteins particularly downregulated in HSV-1 were Peptidyl-prolyl cis-trans isomerase FKBP4 (FKBP4), electron transfer flavoprotein subunit alpha, mitochondrial (ETFA), glutathione peroxidase 1 (GPX1), lysosomal Pro-X carboxypeptidase (PRCP), delta-aminolevulinic acid dehydratase (ALAD), and bile acyl-CoA synthetase (SLC27A5). Upregulated catabolic proteins in HSV-1 were hydroxyacid-oxoacid transhydrogenase, mitochondrial (ADHFE1), short-chain specific acyl-CoA dehydrogenase, mitochondrial (ACADS), and glutathione S-transferase omega-1 (GSTO1).

Our results were consistent with the literature in terms of dysregulated metabolism coinciding with herpes infection. This suggests that our pipeline was effective in identifying host-responses to infection in the CSF. However, there were several notable differences. For example, we identified downregulated metabolic pathways in HSV-1, which has been described previously [15]. Similarly, previous mass spectrometric analysis of HSV-1-infected cells in culture have found dysregulated glycolytic and pentose phosphate pathway intermediates [16,17].

A review [18] emphasizes the interplay of metabolic dysregulation in herpesvirus infection, which also influences host immune response, more specifically, how viral infection by HSV-1 induces mitochondrial stress and increased glycolysis. This mitochondrial stress induces a host antiviral immune response. Mitochondrial stress also induces oxidative stress, which is known to play a role in neurodegeneration. Further, a clinical study [19] showed signs of hypometabolism in parts of the brain in patients with herpes encephalitis when compared to controls. In short, with more evidence, we see how our results of dysregulated metabolism in HSV-1 underscores what is already known in the literature and clinical studies.

Overall, our data has shown how the increase in several metabolic proteins could possibly correlate with the severity of herpesvirus infections.

Our data also showed, in the samples from the HSV-1-infected patients, the pronounced downregulation of five ribosomal proteins in comparison to HSV-2 infections: ribosomal protein L11 (RPL11), L7a (RPL7A), L27a (RPL27A), L34 (RPL34), and L28 (RPL28). Our results are consistent with the literature evidence that, during herpes infection, translation of cellular mRNA transcripts is decreased while viral translation is increased [20,21]. The shut off of host RNA synthesis has been shown as early as 4 h post HSV-1 infection [22].

HSV-1 has a known propensity to highjack and alter host cellular translation machinery [23,24,25]. We observed, in HSV-1-infected patients, the downregulated eukaryotic translation initiation factor 2 subunit 1 (EIF2S1) and Eukaryotic initiation factor 4A-II (EIF4A2). We believe this is the first human CSF study demonstrating directional dysregulation of translation between HSV-1 and HSV-2.

Among the most significant proteins in our 3-way ANOVA analysis were 14-3-3 protein gamma (1433G), 14-3-3 protein epsilon (1433e), 14-3-3 protein eta (1433F), as well as protein S100-A9 (S10A9) and protein S100-A8 (S10A8), which together form calprotectin. Our results consistently show that the aforementioned proteins were most dysregulated in HSV-1 compared to HSV-2 and VZV. For example, 14-3-3 family proteins are involved in the progression and pathogenesis of viral infections [26]. More importantly, HSV-1 has been described as being directly associated with 14-3-3 proteins to induce host apoptosis in order to favor its replication [26,27]. The role of 14-3-3 proteins in apoptosis might explain the aggressive tendencies of HSV-1 infections. Calprotectin, i.e., S100A8 an S100A9, are part of the S100 family of calcium-binding proteins that play an important role in inflammation [28]. One study showed an increase in S100A9 expression in dorsal root ganglia in mice infected with HSV-1 [29], which is consistent with our results that showed an increase in S100A9 in HSV-1-infected patients.

Overall, this concordance between our findings and the literature on these proteins suggests that our methodology may prove robust for the discovery of novel CSF markers to distinguish herpesviruses in the context of CNS infection. We acknowledge that this study had limitations. Foremost, this was a proof of concept study. As such, our primary goal was demonstrating feasibility of CSF as a biofluid that can be used for identifying potential drivers as well as novel functional and mechanistic insight into viral pathogens and related CNS pathologies. Another limitation was our small sample size with variability in the timing of CSF collection and the onset of symptoms. Thus, our findings are not sufficiently robust to constitute biomarker discovery, and instead highlight our method’s recapitulation of known proteins and pathways from the herpes literature as it provides validation of the relevancy of our approach. Larger studies with longitudinal neuropsychological testing and timed CSF collections will be needed to validate findings and determine whether CSF markers such as the 14-3-3 family proteins we found highly elevated in HSV-1 can predict future cognitive impairment. Furthermore, we lack non-CNS involved control samples for our herpes cohorts. This will be an important addition to the next iteration as predictive CSF-based biomarkers that can differentiate between limited mucocutaneous disease versus disease with CNS involvement in herpesvirus infections, which could assist in clinical decision-making regarding diagnostic procedures and initiation of empiric oral or intravenous therapy. While our results were congruent with prior findings from the literature supporting the validity of our proof-of-concept study, an independent and much larger validation cohort study taking into consideration potential confounding factors, such as age and possibly investigating disease across lifespan, is still required to determine if our findings can be used clinically.

In conclusion, our study illuminates that disease-associated changes in the CSF proteome closely resemble molecular changes that have been described in the herpesvirus literature. As such, this study supports the notion that our CSF proteomics method is an appropriate technique for the design and tracking of disease-modifying therapeutics for neurological complications associated with herpes viral infection. Our methodology of characterizing CSF proteomes can reduce a critical barrier in identifying potential targets involved in disease states in a herpes viral infection. In the future, CSF proteomics can be employed to (1) uncover proteins and biological pathways underlying herpesviruses, (2) identify herpesvirus-specific CSF protein signatures and, thus, create a herpesvirus specific biomarker panel, and (3) tailor novel precision medicine approaches for the treatment of serious CNS infection and diseases with lifelong consequences.

## Figures and Tables

**Figure 1 viruses-14-02757-f001:**
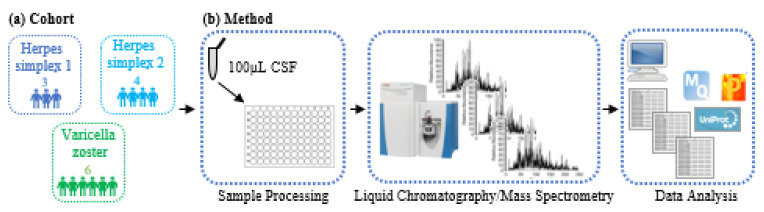
Schematic of the CSF proteomics platform including LC–MS/MS acquisition and data analysis, which was used to process one hundred µL of CSF from thirteen herpes infected patients. (**a**) Cohort schematic comparing three HSV-1-, four HSV-2-, and six VZV-infected patients. (**b**) Method schematic of our in-house-developed high throughput CSF proteomics pipeline.

**Figure 2 viruses-14-02757-f002:**
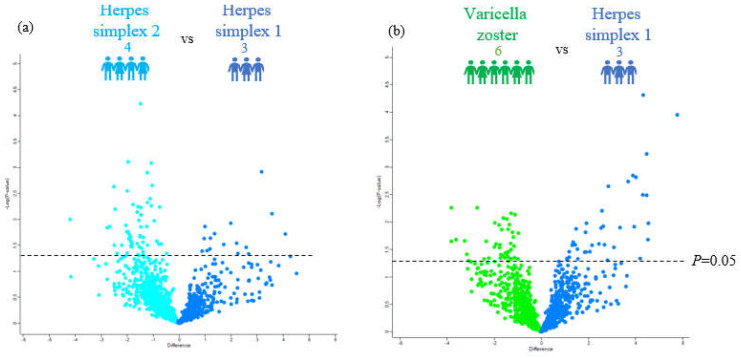
CSF proteome changes in herpes viruses. (**a**) Volcano plot analysis of three HSV-1- (blue) and four HSV-2-infected patients (turquoise) with a *p* value cut off of 0.05 (i.e., equivalent to −log 10 of 1.3). (**b**) Volcano plot analysis of three HSV-1- (blue) six VZV-infected patients (green) with a *p* value cut off of 0.05 (i.e., equivalent to −log 10 of 1.3).

**Figure 3 viruses-14-02757-f003:**
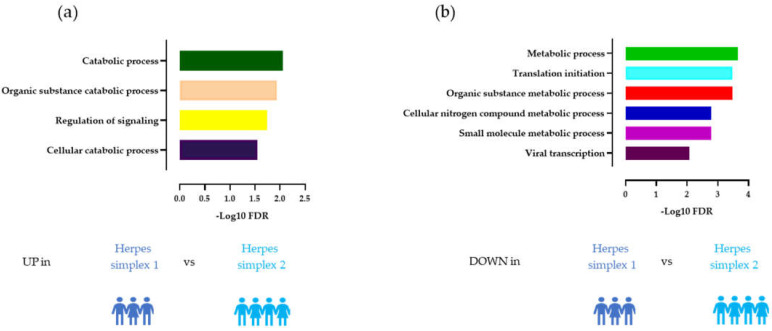
Biological pathway analysis indicating which pathways are enriched in herpes subtypes. (**a**) Biological pathways upregulated in HSV-1 vs. HSV-2 infection. (**b**) Biological pathways and interacting proteins downregulated in HSV-1 vs. HSV-2 infection.

**Figure 4 viruses-14-02757-f004:**
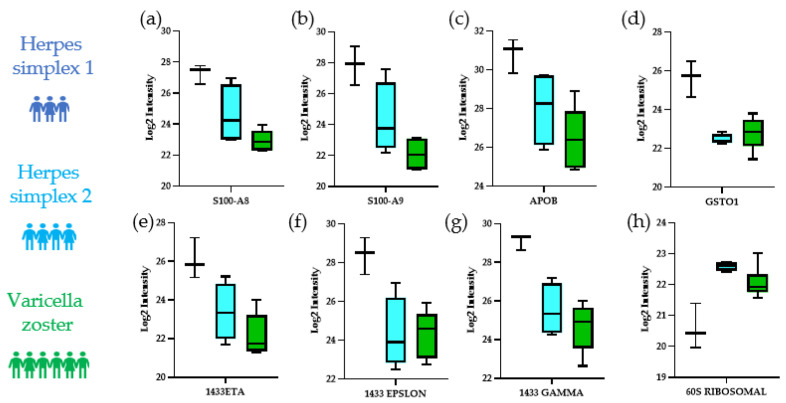
Boxplot analysis of ANOVA significant proteins between HSV-1, HSV-2, and VZV. Log2 intensity values are plotted. (**a**) Protein S100-A8 (S10A8) (*p* = 1.50 × 10^−3^); (**b**) Protein S100-A9 (S10A9) (*p* = 1.20 × 10^−3^); (**c**) Apolipoprotein B-100 (APOB) (*p* = 1.04 × 10^−2^); (**d**) Glutathione S-transferase omega-1 (GSTO1) (*p* = 4.00 × 10^−4^); (**e**) 14-3-3 protein eta (1433F) (*p* = 4.00 × 10^−3^); (**f**) 14-3-3 protein epsilon (1433E) (*p* = 5.00 × 10^−3^); (**g**) 14-3-3 protein gamma (1433G) (*p* = 1.00 × 10^−4^); (**h**) 60S ribosomal protein L34 (RL34) (*p* = 1.00 × 10^−4^).

**Figure 5 viruses-14-02757-f005:**
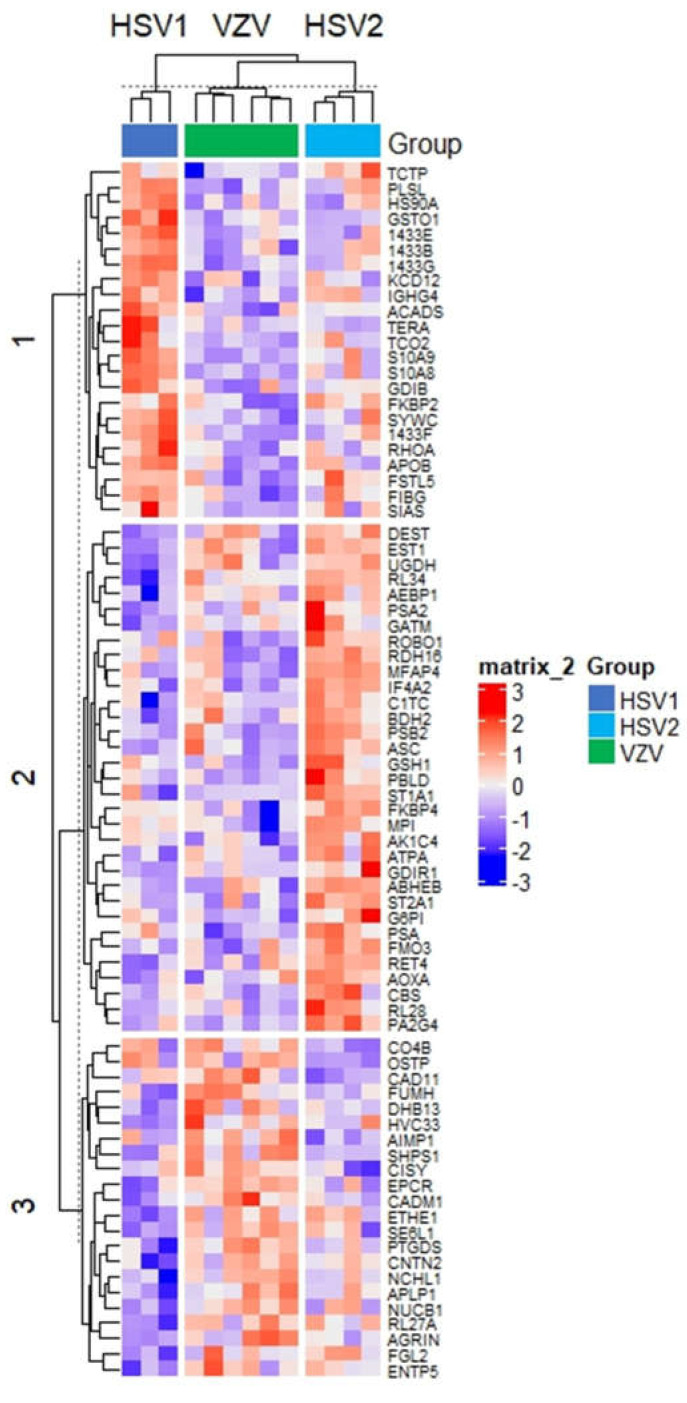
Hierarchical clustering of 78 ANOVA significant proteins comparing HSV-1, HSV-2, and VZV. Red indicates an increase in log2 intensity while blue indicates a decrease in log2 intensity.

**Table 1 viruses-14-02757-t001:** Characteristics of Cohort.

Pathology	Age	Sex	Race	PMH	Reason for Lumbar Puncture	Relevant Diagnostic Testing
HSV-1	58	M	Black	Prostate adenocarcinoma, history of pulmonary tuberculosis and syphilis	Altered mental status	CSF PCR: HSV-1: positive, HSV-2: negative, VZV: negative; CSF Gram stain: no organisms; Bacterial culture: no growth; CSF RT-QuIC or 14-3-3: not tested
HSV-1	25	M	White	Dental abscesses	Altered mental status, fever, seizure	CSF PCR: HSV-1: positive, HSV-2: negative, VZV: not tested; CSF Gram stain: no organisms; Bacterial culture: no growth; CSF RT-QuIC or 14-3-3: not tested
HSV-1	57	M	White	Chronic back pain	Altered mental status	CSF PCR: HSV-1: positive, HSV-2: negative, VZV: not tested; CSF Gram stain: no organisms; Bacterial culture: no growth; CSF RT-QuIC or 14-3-3: not tested
HSV-2	87	F	White	Hypertension, hyperlipidemia, peripheral arterial disease, peripheral neuropathy	Altered mental status	CSF PCR: HSV-1: negative, HSV-2: positive, VZV: not tested; CSF Gram stain: no organisms; Bacterial culture: no growth; Skin vesicle VZV DFA: negative; CSF RT-QuIC or 14-3-3: not tested
HSV-2	85	M	White	Coronary artery disease, chronic obstructive pulmonary disease, chronic kidney disease, suspected myasthenia gravis s/p thymectomy	Seizure, somnolence, bilateral arm twitching	CSF PCR: HSV-1: negative, HSV-2: positive, VZV: not tested; CSF Gram stain: no organisms; Bacterial culture: no growth; CSF RT-QuIC or 14-3-3: not tested
HSV-2	39	F	White	Hypothyroidism, recurrent aseptic meningitis, migraine without aura	Headache, photophobia/phonophobia, nausea, neck stiffness	CSF PCR: HSV-1: negative, HSV-2: positive, VZV: negative; CSF total VZV antibody (ACIF): <1:2; CSF Gram stain: no organisms; Bacterial culture: no growth; Blood HSV-1 IgG negative, HSV-2 IgG positive; CSF RT-QuIC or 14-3-3: not tested
HSV-2	54	F	White	Reactive airway disease, obstructive sleep apnea, depression and anxiety, remote migraines, history of HSV-2 meningitis	Worsening headache, neck pain, photophobia, diffuse weakness	CSF PCR: HSV-1: negative, HSV-2: positive, VZV: negative; CSF Gram stain: no organisms; Bacterial culture: no growth; CSF RT-QuIC or 14-3-3: not tested
VZV	39	F	White	Depression	Fever, headache, neck stiffness and right flank rash/abdominal	CSF PCR: HSV-1: negative, HSV-2: negative, VZV: positive; CSF Gram stain: no organisms; Bacterial culture: no growth; CSF RT-QuIC or 14-3-3: not tested
VZV	74	F	White	Chronic obstructive pulmonary disease	Right arm/shoulder/back rash, progressive somnolence, event concerning for GTC	CSF PCR: HSV-1: negative, HSV-2: negative, VZV: positive; CSF total VZV antibody (ACIF): 1:32 CSF; Gram stain: no organisms; Bacterial culture: no growth; CSF RT-QuIC or 14-3-3: not tested
VZV	82	F	White	Lung and bladder cancer, atrial fibrillation, schizoaffective disorder	Altered mental status, generalized weakness	CSF PCR: HSV-1: negative, HSV-2: positive, VZV: negative; CSF total VZV antibody(ACIF): 1:128; CSF Gram stain: no organisms; Bacterial culture: no growth; CSF RT-QuIC or 14-3-3: not tested
VZV	78	M	White	Diabetes mellitus, hypertension, hyperlipidemia, coronary artery disease, chronic obstructive pulmonary disease, trigeminal neuralgia	Scalp pain, dysphagia, ataxia, and dizziness	CSF PCR: HSV-1: negative, HSV-2: negative, VZV: negative; CSF VZV total antibody (ACIF): 1:8; CSF Gram stain: no organisms; Bacterial culture: no growth; CSF RT-QuIC or 14-3-3: not tested; Skin vesicle VZV DFA: positive
VZV	79	M	White	Chronic lymphocytic leukemia, chronic obstructive pulmonary disease, interstitial lung disease, chronic kidney disease	Left ptosis, ophthalmoparesis	CSF PCR: HSV-1: negative, HSV-2: negative, VZV: negative; CSF VZV total antibody (ACIF): 1:8; CSF Gram stain: no organisms; Bacterial culture: no growth; CSF RT-QuIC or 14-3-3: not tested
VZV	85	M	White	Atrial fibrillation, hypertension, history of resected benign intracranial neoplasm (per report)	Altered mental status and recent history of cutaneous zoster	CSF PCR: HSV-1: negative, HSV-2: negative, VZV: negative; CSF VZV total antibody (ACIF): 1:16; CSF Gram stain: no organisms; Bacterial culture: no growth; CSF RT-QuIC or 14-3-3: not tested

Based on Provider Notes on the Clinical Examination Prior to Lumbar Puncture. Abbreviations: CSF, cerebrospinal fluid; PMH, past medical history; PCR, polymerase chain reaction; DFA, Direct immunofluorescence; ACIF, anticomplement immunofluorescence. Normal range: PCR = negative; DFA = negative; ACIF < 1:2.

## Data Availability

Not applicable.

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
