# Peer review of "Using CSF Proteomics to Investigate Herpesvirus Infections of the Central Nervous System"

_viruses, 2022, doi:10.3390/v14122757_

Round 1

Reviewer 1 Report (New Reviewer)

Authors succesfullyed show the feasibility of developing CNS proteomic signatures of the host response in alpha herpes infections. The study is an excellent example for investigating the pathophysiology driving virus-associated neurological disorders, developing biomarkers of morbidity, and generating personalized therapeutic strategies in near future.

Revisions were clearly adressed to clarify the missing points.

Reviewer 2 Report (New Reviewer)

Ahmed et al., report proteomic analysis of 100 µl size samples of CSF from patients with nervous system infections with HSV-1, HSV-2 and VZV and show some possibly interesting differences protein composition between virus infections.  The results reported are potentially significant in that they represent a first step in identification of biomarkers for specific viral infections of the central nervous system and an interesting approach to understanding host responses to viral infection.  Unfortunately, it is difficult to tell from the results as they are reported here exactly how promising this approach is.

The authors state that their proteomic pipeline identified a total of 1231 proteins, but do not indicate their criterion for identification.  Is a single peptide enough to constitute an identification? If so, OK, but this will create issues with the statistical analyses used and the authors need to address more completely how these analyses were performed.  For example, if there are many proteins represented by a single peptide, then the authors must be using imputed values for missing peptides in their statistical analyses.  No description is given of how imputation is performed and how many values are imputed in their analyses.  As such, it is difficult to evaluate the reliability of their analysis.  The authors should present a summary of their proteomic results giving some indication of the peptide representation in their identified proteins and more detail about how the statistical analyses were performed.

The value of the comparisons between infections with different viruses are also difficult to assess for several reasons.  The first of these (that there are no uninfected controls) is acknowledged by the authors in the discussion.  Obviously, this study would be much stronger with such controls, but there is little harm in the authors speculations on the possible biological significance of differences that they do observe.  A much more serious problem is that there is no independent validation (by immunoblot, ELISA, or other techniques) of the possibly interesting differences that they see in the proteomic analyses.  Given the small sample sizes available, this may be infeasible, but without knowing more about the proteomic analysis (see comments in the previous paragraph), the reader is left with no idea how robust these results are.

Reviewer 3 Report (New Reviewer)

This resubmission was a revised manuscript to address concerns from a previous round of review.  Regarding the previous review comments, the manuscript reasonably stands as a proof-of-concept study and the additional text emphasizing this was a good addition.  Furthermore, the expanded discussion adds additional studies and dialogue on the dysregulated pathways.  While additional sample sizes and larger cohorts are still to be desired for more conclusive findings, the authors have adequately addressed the concerns in their letter and revised manuscript.   I have a few additional text corrections that might be helpful:

Title:  I would say either "Herpesvirus Infections" or "Herpesviruses of the Central Nervous System" but not the double plural

Italicize et al. throughout the document Line 109: hyphenate "in house" Line 111: place a space in "in1" Table 1:  There is an inconsistent use of quotes between rows for the text in the "Relevant Diagnostic Testing" column Section 3.2:  I would encourage the use of capital E in this section for scientific notation vs lowercase e, as is done below Lines 196 and 198: hyphenate HSV1 Figure 4:  the second parentheses for (h) is missing Line 245: change convenience to convenient Line 286: change increase to increased Line 291: hyphenate "HSV-1-infected" Line 293:  This study indicates, rather than demonstrates a dysregulation of translation.  Additional molecular work would be required for a proper demonstration

This manuscript is a resubmission of an earlier submission. The following is a list of the peer review reports and author responses from that submission.

Round 1

Reviewer 1 Report

The authors describe the feasibility of proteomic method for the analysis of CSF samples from patents with alphaherpesvrus infection of the CNS.  I have some questions and comments:

-          The authors report the results regarding 13 CSF samples, which is maybe too low to draw any conclusions

-          In my opinion, the analysis of control CSF from patients with non-infectious CNS disease is lacking

-          Authors should make some assumptions to explain the modifications of metabolic pathways observed in this work

-          Spelling mistakes to correct: human herpes simplex virus 1 (instead of type 1); human herpesvirus (instead of herpes virus)

Reviewer 2 Report

This work describes the proteomic analysis of human cerebrospinal fluid (CSF) of patients with meningitis or encephalitis due to Varicella-zoster virus (VZV), or human herpes simplex type 1 or 2 (HSV-1, HSV-2 respectively). The work describes specific proteins identified in HSV-1 patients that are more abundant when compared to HSV-2 or VZV patients. Additionally, this work describes specific metabolic pathways that are down regulated as a result of HSV-1 disease specifically. The authors propose that this work will be useful in biomarker discovery and personalized therapeutic intervention. Overall, the manuscript is well written, and I would accept pending the authors address my concerns outlined below.

11.       Patients have multiple other bacterial infections. These can be a source of elevated CSF proteins. For example calprotectin is considered a biomarker of acute bacterial respiratory infections. Its highly elevated in bacterial infection compared to viral. (Aleksandra Havelka, Scientific reports, 2020).

22.       Age group of HSV-1 positive patients is very young compared to the ones with HSV-2 and VZV. Does it affect the outcome?

33.       Elevated 14-3-3 protein is a diagnostic of CJD disease, hereditary or acquired. Are these patients tested for CJD?

44.       HSV-1, HSV-2, VZV positive group patients have different background illnesses (table-1), so the conditions are not uniform. How specific are these elevated CSF proteins to herpes infection? Is it possible to conclude?

55.       The patients who were selected for one viral infection positive only, are they negative for the remaining two?  

66.       All patients were taking acyclovir during the lumber puncture. Why? Did they show any HSV symptoms? So there was no active replication of herpes at that time due to treatment. Mostly HSV remains latent in the brain.

77.       Did you have any control CSF samples who is only herpes positive but does not have other underlying issues? Can you consider that?

88.       Figure 3 a and 3 b- very unclear pathway (ball and stick images). The font needs to be enlarged or the resolution could be better.

99.       How were the meningitis/encephalitis patients tested for herpes? Serological test? Most population will be seropositive for HSV. Were they negative for respective bacterial candidates? (meningitis can be caused by many bacteria as well).

110.   As a report on human CSF samples this is a good analysis but the conclusions made from the data, considering the variations of samples (variable sample number, age, sex, underlying infections etc), is not solid.